# Chemical Synthesis, Pharmacokinetic Properties and Biological Effects of JM-00266, a Putative Non-Brain Penetrant Cannabinoid Receptor 1 Inverse Agonist

**DOI:** 10.3390/ijms23062923

**Published:** 2022-03-08

**Authors:** Tania Muller, Laurent Demizieux, Stéphanie Troy-Fioramonti, Chloé Buch, Julia Leemput, Christine Belloir, Jean-Paul Pais de Barros, Tony Jourdan, Patricia Passilly-Degrace, Xavier Fioramonti, Anne-Marie Le Bon, Bruno Vergès, Jean-Michel Robert, Pascal Degrace

**Affiliations:** 1Equipe Physiopathologie des dyslipidémies, Unité Mixte de Recherche Université de Bourgogne Franche-Comté - Institut National de la Santé et de la Recherche Médicale (UMR-INSERM) 1231, F-21000 Dijon, France; tania.muller67@gmail.com (T.M.); laurent.demizieux@u-bourgogne.fr (L.D.); stephanie.fioramonti@hotmail.fr (S.T.-F.); chloejeanne.buch@gmail.com (C.B.); julia.leemput@u-bourgogne.fr (J.L.); tony.jourdan@u-bourgogne.fr (T.J.); patricia.degrace@u-bourgogne.fr (P.P.-D.); bruno.verges@u-bourgogne.fr (B.V.); 2Centre des Sciences du Goût et de l’Alimentation (CSGA), AgroSup Dijon, Centre National de la Recherche Scientifique (CNRS), Institut National de Recherche pour l’Agriculture, l’Alimentation et l’Environnement (INRAE), Université de Bourgogne Franche-Comté, F-21000 Dijon, France; christine.belloir@inrae.fr (C.B.); xavier.fioramonti@inrae.fr (X.F.); anne-marie.le-bon@inrae.fr (A.-M.L.B.); 3Plateforme de Lipidomique, Unité Mixte de Recherche Université de Bourgogne Franche-Comté - Institut National de la Santé et de la Recherche Médicale (UMR-INSERM) 1231, F-21000 Dijon, France; jppais@u-bourgogne.fr; 4Unité de Nutrition et Neurobiologie Intégrative (NutriNeuro), Unité Mixte de Recherche Université de Bordeaux - Institut National de Recherche pour l’Agriculture, l’Alimentation et l’Environnement (UMR-INRAE) 1286, F-33000 Bordeaux, France; 5Unité de Recherche Cibles et Médicaments des Infections et de l’Immunité (UR115 IICiMed), Institut de Recherche en Santé 2 Nantes Université, F-44200 Nantes, France; jean-michel.robert@univ-nantes.fr

**Keywords:** endocannabinoid system, CB1R antagonist, Rimonabant, SWISSADME prediction, drug discovery, pharmacokinetics, obesity

## Abstract

Targeting cannabinoid 1 receptors (CB1R) with peripherally restricted antagonists (or inverse agonists) shows promise to improve metabolic disorders associated with obesity. In this context, we designed and synthetized JM-00266, a new CB1R blocker with limited blood–brain barrier (BBB) permeability. Pharmacokinetics were tested with SwissADME and in vivo in rodents after oral and intraperitoneal administration of JM-00266 in comparison with Rimonabant. In silico predictions indicated JM-00266 is a non-brain penetrant compound and this was confirmed by brain/plasma ratios and brain uptake index values. JM-00266 had no impact on food intake, anxiety-related behavior and body temperature suggesting an absence of central activity. cAMP assays performed in CB1R-transfected HEK293T/17 cells showed that the drug exhibited inverse agonist activity on CB1R. In addition, JM-00266 counteracted anandamide-induced gastroparesis indicating substantial peripheral activity. Acute administration of JM-00266 also improved glucose tolerance and insulin sensitivity in wild-type mice, but not in CB1R^−/−^ mice. Furthermore, the accumulation of JM-00266 in adipose tissue was associated with an increase in lipolysis. In conclusion, JM-00266 or derivatives can be predicted as a new candidate for modulating peripheral endocannabinoid activity and improving obesity-related metabolic disorders.

## 1. Introduction

The endocannabinoid system (ECS) is comprised of cannabinoid receptors, their endogenous ligands (endocannabinoids) and proteins responsible for their synthesis and degradation. Cannabinoid-1 receptor (CB1R) is highly expressed in the brain where it plays key roles in the control of nociception, appetite, motor activity, mood, and memory [1]. However, CB1R is also present in peripheral organs involved in the control of energy metabolism, including the liver, intestine, pancreas, muscle and adipose tissue [2]. Hence, this receptor emerged as a critical regulator of lipid metabolism, gastrointestinal motility, or cardiovascular function [3]. Evidence has also accumulated that obesity is associated with an overactivation of the ECS [4,5,6]. The identification of a role for ECS in the regulation of food intake and energy metabolism suggested therapeutic strategies aimed at blocking CB1R to treat obesity and related disorders. As such, Rimonabant (SR141716) was the first CB1R specific inverse agonist approved for therapeutic purposes and commercialized by Sanofi in Europe [7]. Obese and overweight patients treated with Rimonabant showed significant weight reduction and improvement of many cardio-metabolic parameters [8,9,10]. However, because of its psychiatric side effects, such as depression, anxiety, sleep disturbances and suicide risk, it was withdrawn from the European market in 2008, drawing into question the safety of first-generation antagonists [11,12]. Nevertheless, many studies showed that food intake reduction was not the only mechanism responsible for weight reduction in diet-induced obese mice treated with Rimonabant, thus, highlighting a role for peripheral CB1R in the regulation of energy metabolism [2,13]. Hence, the strategy consisting in targeting peripheral CB1R with no-brain-penetrant antagonists to improve metabolic disorders associated with obesity has gained growing interest [14].

Among the various second-generation antagonists, AM6545, TM38837, JD5037 and MRI-1891 (renamed INV-202) exhibited beneficial effects on body weight and metabolism in pre-clinical trials in mice and rats [15,16,17,18]. However, to date, only the antagonist TM38837 developed by 7TM Pharma and INV-202 developed by Inversago Pharma successfully completed phase I clinical trial [19,20].

Here, we designed a new CB1R blocker, JM-00266, modifying the structure of Rimonabant to limit blood–brain barrier crossing. Pharmacokinetic studies revealed that JM-00266 exhibits CB1R inverse agonist properties and reduced brain penetration compared to Rimonabant. In addition, behavioral experiments revealed no central activity of JM-00266 suggesting that it could be a potential candidate for selectively targeting peripheral CB1R. We also showed that acute administration of this compound to mice improves glucose tolerance, insulin sensitivity and increases adipose tissue lipolysis.

## 2. Results

### 2.1. Structure, Pharmacological Properties and Bioavailability of JM-00266

#### 2.1.1. Synthesis and Structure of JM-00266 and JM-00252

JM-00266 (1,4-di-(4-méthylthiophenyl)-3-phthaloylazetidin-2-one), is a compound possessing a ß-lactam (azetidin-2-one) scaffold 1,4-diaryl substituted with a protected (phthaloylated) amine function on the 3 position of the four-membered ring. Theoretically, the synthesis procedure leads to two diastereoisomers (trans and cis) but in this case, only the compound trans was obtained (Figure 1A). However, with the ring being constituted of two asymmetric carbons, the compound JM-00266 exists as the racemic of the two enantiomers (3S, 4S) and (3R, 4R). In the present work, we used the racemic form. The structure of JM-00252, synthesized as described for JM-00266 with appropriate intermediates and used as an internal standard for LC-MS/MS analysis, is presented in Appendix A.

#### 2.1.2. Tissue Distribution and Brain Penetrance

JM-00266 was designed to display very low amphiphilic properties in order to reduce its brain permeability compared to Rimonabant used as a reference compound. Thus, Topological Polar Surface Area (TPSA) value obtained by the Swiss ADME server was far higher for JM-00266 than for Rimonabant (108.29 vs. 50.16 Å^2^, respectively; Figure 1A). In addition, a high gastrointestinal absorption is predicted for JM-00266 as displayed in the white region of the boiled-egg view (Figure 1B). In vivo pharmacokinetics experiments were carried out to determine JM-00266 bioavailability and transport across the BBB. We first analyzed the plasma levels of JM-00266 after oral administration of the drug dissolved in oil or after i.p. injection of a DMSO/tween solution. Drug plasma concentration reached peak levels at 30 min and 1 h after i.p. and oral dosing, respectively (Figure 2A). Interestingly, for the same dose, the maximal drug plasma concentration was higher after i.p. than oral administration (378 ± 27 vs. 203 ± 18 ng·mL^−1^). Conversely, the same experiment repeated with Rimonabant indicated that the bioavailability of the drug was almost 2-fold higher after oral than i.p. administration (Figure 2B). Data also show that whatever the administration route, the bioavailability of Rimonabant is far higher than that of JM-00266. When administered intravenously, drug kinetic patterns indicate a slightly higher clearance for JM-00266 than Rimonabant (Figure 2C).

Then, tissue distribution of JM-00266 and Rimonabant was determined 4h after gavage of drugs prepared in oil. We first observed that both drugs preferentially accumulated in the liver and the adipose tissue whereas brain concentration remained very low (Figure 2D). While lipophilicity of JM-00266 could account for adipose tissue accumulation, the poor concentration in the brain should be considered as the first indication of its low capacity to cross BBB. It is also noteworthy that JM-00266 was recovered at lower concentrations in plasma and tissues than Rimonabant, in accordance with its lower bioavailability observed in kinetic studies.

Since JM-00266 was designed to present limited central nervous system (CNS) penetration, we determined brain/plasma ratios (B/P) and brain uptake index (BUI) of JM-00266 in comparison with Rimonabant, known to penetrate the brain. For B/P measurements, JM-00266 and Rimonabant were prepared in DMSO/tween 80 aqueous solutions in order to favor rapid appearance in blood circulation and limit variations related to prolonged exposure. Whatever the administration route (i.p. or oral), B/P was significantly lower for JM-00266 than Rimonabant (Figure 2E). It is worth noting that the administration route strongly affected the distribution of Rimonabant in the brain and plasma while that of JM-00266 was not modified.

To further study BBB permeability excluding variations due to administration route and dilution of the compound in peripheral vascularization and organs, we performed a single co-injection of JM-00266 and Rimonabant in the carotid artery of rats. The BUI index for JM-00266 was calculated using the BUI of Rimonabant as a reference. The mean BUI was 7.8% ± 0.6 indicating a low ability for JM-00266 to cross BBB compared to Rimonabant (Figure 2F).

### 2.2. cAMP Functional Assay

Activation of CB1R is known to decrease intracellular levels of cAMP by inhibiting adenylate cyclase activity primarily through G_i/o_ signaling [21]. Conversely, an inverse agonist will induce the opposite effect, thus leading to an increase in cAMP production [22]. Using HEK293T/17 cells transfected with murine CB1R, we showed that treatment with JM-00266 alone induced a dose-dependent increase in cAMP formation (EC_50_ = 248.6 µM) indicating that the compound behaves as an inverse agonist for the receptor (Figure 3A). This response was lost in the presence of pertussis toxin treatment, which prevents Gi/o coupling to CB1R and with increasing concentrations of the CB1R agonist anandamide. However, it should be noted that cAMP level elevation was stronger with Rimonabant than JM-00266 for the same concentrations tested (EC50 Rimonabant = 89.2 nM) suggesting a lower potency of JM-00266 than Rimonabant (Figure 3B).

### 2.3. Effect of JM-00266 on Brain-Mediated Functions and Gastro-Intestinal Transit

We next investigated the potential impact of JM-00266 on the brain using behavioral tests based on the ability of CB1R agonists or antagonists to induce changes in food intake, body temperature, or anxiety-like behavior [1].

#### 2.3.1. Food Intake

Effects of JM-00266 and Rimonabant on feeding behavior were recorded for 3 h in animals deprived of food for 24 h and subjected to a single i.p. administration of the selective CB1R agonist ACEA. As reported previously in rats [23], acute CB1R activation had no effect on food intake in food-deprived mice (Figure 4A). In contrast, co-administration of Rimonabant with ACEA elicited a strong hypophagia that was not observed with JM-00266 (Figure 4A).

#### 2.3.2. Body Temperature

Body temperature was first monitored after a single injection of anandamide in the presence or not of JM-00266. Data indicated that JM-00266 was not able to counteract the decrease in rectal temperature induced by anandamide (Figure 4B). Since anandamide induced hypothermia has been reported to depend on both central CB1R and peripheral TRPV1 [24,25], we repeated the experiment targeting CB1R using the selective agonist ACEA and the selective antagonist Rimonabant. In these conditions, pre-treatment of mice with Rimonabant fully inhibited the hypothermic response to ACEA whereas JM-00266 had no effect (Figure 4C) suggesting that the drug did not compete with ACEA on brain CB1R.

#### 2.3.3. Anxiety-like Behavior

Rimonabant did not significantly alter the locomotor activity or the number of center entries, while it significantly decreased the time spent in this area. By contrast, JM-00266 did not alter these 3 parameters (Figure 4D). Taken together, these findings suggest that JM-00266, in contrast to Rimonabant, does not promote anxiety-like behavior which confirms its limited impact on the CNS.

#### 2.3.4. Gastro-Intestinal Transit

The current literature indicates that CB1R activation inhibits gastrointestinal motility [26,27]. In this context, we tested the capacity of JM-00266 to counteract gastroparesis induced by acute anandamide injection in mice. Our results indicated that the slowing down of gastro-intestinal transit that followed anandamide administration was abrogated by a pre-treatment with JM-00266 suggesting that this compound was indeed able to inactivate peripheral CB1R (Figure 4E).

### 2.4. Metabolic Effects of JM-00266

#### 2.4.1. Glucose Tolerance

OGTT revealed that acute JM-00266 treatment improved glucose tolerance compared to vehicle treatment in wild-type mice (Figure 5A). During OGTT, plasma insulin levels of mice administered JM-00266 were lower than vehicle (Figure 5B). In parallel, during ITT, mice subjected to JM-00266 injection showed lower glycemia over time compared to vehicle treated mice, suggesting an improvement of insulin sensitivity (Figure 5C). Interestingly, JM-00266 failed to improve glucose tolerance in CB1R^−/−^ mice, indicating a CB1R-dependent mechanism (Figure 5D).

#### 2.4.2. Adipose Tissue Lipolysis

The fact that JM-00266 accumulates in fat depots (Figure 2D) prompted us to measure the impact of the drug on adipose tissue lipolysis. Interestingly, ß-adrenergic-induced glycerol production was increased by JM-00266 treatment compared with vehicle (Figure 6A). In addition, when lipolysis was measured in vitro, using visceral adipose tissue explants collected from mice injected with JM-00266 4 h before sacrifice, glycerol release was also higher than in the control (Figure 6B).

## 3. Discussion

Accumulating evidence indicates that the inactivation of peripheral CB1R represents a promising therapeutic strategy to control obesity and related metabolic disorders. Identification of CB1R blockers devoid of psychological effects becomes a major challenge for the scientific community [14]. In this report, we describe a new compound, JM-00266, designed from the specific CB1R inverse agonist Rimonabant template incorporating polar functionality.

The CB1R is a G-protein-coupled receptor linked to different signal transduction pathways including inhibiting adenylyl cyclase activity [21]. Using a functional cAMP assay to monitor G protein-coupled receptor activity, we provide evidence that JM-00266 can indeed raise cAMP levels in HEK293 cells overexpressing CB1R indicating that this compound behaves as an inverse agonist. In addition, the fact that pertussis toxin or anandamide treatment abrogated the effect of JM-00266 on cAMP levels further suggests a CB1R-dependent action. While the EC_50_ values indicate a far less potency for JM-00266 than for Rimonabant, it is noteworthy that JM-00266 was able to fully abrogate the slowing down of gastrointestinal transit induced by anandamide supporting the existence of a substantial action of the drug on CB1R.

The bioavailability of JM-00266 was first examined by TPSA, a medicinal chemistry metric commonly estimating passive molecular transport through membranes [28]. Molecules with a TPSA > 120 Å^2^ usually poorly cross cell membranes and would generally display low oral bioavailability while molecules with TPSA < 70 freely penetrate BBB and thus potentially interact with receptors in the central nervous system [29]. The high TPSA value of JM-00266 likely accounts for its weaker peripheral bioavailability compared to Rimonabant but it also predicts that JM-00266 is a low brain-penetrant compound. This was confirmed by pharmacokinetics parameters related to BBB transport, such as B/P ratio and BUI. We showed that B/P ratios for JM-00266 were always comprised between 0.3 and 0.5 whatever the administration route tested while that of Rimonabant were >1. According to the literature, a compound with a B/P ratio greater than 0.3–0.5 is considered to have sufficient access to the central nervous system while compounds with B/P ratios greater than 1 freely cross the BBB [30]. Hence, the present results suggest that the ability of JM-00266 to access to the central nervous system is much lower than Rimonabant. The low rate of transport of JM-00266 across BBB was further supported by the weak BUI measured when the drug was directly injected into the brain (7.8% of the value of Rimonabant used as reference).

The ability of drugs to cross the BBB cannot be directly linked to their CNS activity. So, as we observed that part of JM-00266 might access the brain, we also estimated its potential central activity performing some behavioral tests commonly used to screen drugs inducing cannabinoid-mediated effects. Administration of Rimonabant at 10 mg/kg is commonly used in studies dealing with its peripheral impact on metabolism in mice [13,31,32,33,34,35]. In addition, this dose was shown to induce brain CB1R-dependent behavioral effects [36,37]. Accordingly, we applied a similar dose for the determination of JM-00266 pharmacokinetics parameters in comparison with Rimonabant. However, since our data analysis revealed a lower potency of JM-00266 compared to Rimonabant, we used a higher dose of JM-00266 (20 mg/kg) for behavioral and metabolic studies.

The observation that acute administration of JM-00266 did not induce hyperphagia, hypothermia, or changes in anxiety behavior while Rimonabant modified these parameters suggests that JM-00266 does not bind brain CB1R. Convergent studies indicate that activation of brain CB1R stimulates food intake [7,38,39]. Hence, it is interesting to note that JM-00266 did not modify the amount of food ingested during refeeding as Rimonabant did. In rodents, it is well characterized that activation of CB1R by administration of agonists induces hypothermia [40,41] and that this effect is dependent on central CB1R. The fact that JM-00266 was not able to counteract the decrease in core body temperature induced by ACEA as Rimonabant did, further illustrates its lack of central activity. In addition, contrary to agonists, CB1R antagonists induce anxiogenic-like effects in laboratory animals [36,42]. Previous studies report modifications of locomotor activity and anxiety-related behavior following Rimonabant administration at 10mg/kg in rodents [43,44]. In the open-field exploration test, mice treated with Rimonabant spent less time exploring the central area of the chamber compared to wild type suggesting anxiety-related behavior which was not observed with JM-00266.

Accumulating evidence indicates that peripheral CB1R blockade improves lipid and carbohydrate homeostasis acting simultaneously on several peripheral organs [3,45]. Thus, activation of hepatocyte CB1R has been shown to induce liver lipogenesis and decrease ß-oxidation [46,47]. Conversely, CB1R blockage or hepatocyte-specific deletion protects mice fed a high fat diet from insulin resistance [48,49]. In the same way, our data showed that blockade of peripheral CB1R by JM-00266 is associated to better utilization of glucose by peripheral tissues as suggested by the results of OGTT and ITT.

As JM-00266 is markedly distributed into fat tissues, we investigated its possible role on adipocyte lipid metabolism. The current literature strongly supports the notion that obesity induces overactivation of the ECS in adipose tissue in both animals and humans [4,6,33,50,51]. Further, an increase in the endocannabinoid tone in adipose tissue is associated with a dysregulation of adipocyte metabolism, thus, promoting fat mass expansion and related disorders [52,53,54,55,56]. In line with this, we recently indicated that activation of ECS in adipose tissue decreased lipolysis [57,58]. Here, we interestingly observed that JM-00266 treatment potentiates ß3 adrenergic receptor-stimulated lipolysis in vivo suggesting this compound might improve fat mobilization targeting adipose tissue CB1R. In vitro studies also indicated that glycerol release by adipose tissue explants collected from mice previously treated with JM-00266 was greater than in control suggesting that JM-00266 could increase fatty acid mobilization in vivo after being incorporated into fat tissues.

In conclusion, despite its low bioavailability and its weak CB1R binding capacities (as described with the cAMP Glosensor functional test), JM-00266 showed marked CB1R-dependent in vivo effects. Future directions should investigate different formulations, such as the preparation of liposomes or nano-emulsions to increase solubilization and delivery efficiency. According to its low brain penetrance and to its CB1R inverse agonist properties, we further believe that the chemical structure of JM-00266 may serve as a template to design derivatives that might become of great interest as new drugs for modulating peripheral ECS tone and improving metabolic disorders associated with obesity.

## 4. Materials and Methods

### 4.1. Chemistry

The 1,4-diphenyl-3-phtaloylazetidin-2-ones (*N*-phtaloyl-4,4-diaryl-β-lactams) compounds described in this work (JM-00252 and JM-00266) were synthesized as described in [59] using a Staudinger reaction with a [2+2] ketene-imine cycloaddition leading to the formation to the *trans*-β-lactam formation [60]. The imine compounds were obtained by condensation of appropriate benzylamines on benzaldehydes. The ketene was generated in situ by the action of trimethylamine on the activated compound resulting in the action of phenyl dichlorophosphate used as a *N*-phtaloylglycine activator.

#### 4.1.1. Synthesis of 4-Méthylthiobenzyl-4-méthylthiobenzaldimine

4-methylthiobenzaldehyde (5.57 g, 36.6 mM) and 4-methylthioaniline (5.0 g, 35.92 mM) were refluxed in toluene (50 mL) in the presence of anhydrous sodium sulfate for 4 h. Then, the solvent was evaporated under a vacuum. The crude solid was washed with diisopropyle oxide (2 × 20 mL) and the suspension was obtained, and filtered to give 7.66 g (78%) of the imine **1**. This compound was used without further purification.

MP °C = 143–144 (Diisopropyle oxide).^1^H-RMN (CDCl_3_): δ 2.51, s, 3H, 4-SCH_3_; 2.54, s, 3H, 4′-SCH_3_; 7.18, d, 2H, H^3^H^5^, *J* = 6.7 Hz; 7.29, d, 2H, H^2^H^6^; 7.30, d, 2H, H^3′^H^5′^, *J* = 8.4 Hz; 7.80, d, 2H, H^2′^H^6′^; 8.41, s, 1H, HC=N.SM (ESI) *m*/*z* (%): 274 [M + H]^+^IR (KBr, cm^−1^): 1552.53 (ν C=N).

#### 4.1.2. Synthesis of 1,4-Di-(4-Méthylthiophenyl)-3-Phtaloylazetidin-2-one (JM-00266)

Imine **1** (1.38 g, 5 mM), triethylamine (3 mL) and *N*-phtaloylglycine (1.128 g, 5.42 mM) were dissolved in dry dichloromethane (20 mL). The mixture was cooled to 0 °C, and phenyl dichlorophosphate (2.11 g, 10 mM) was added dropwise. Then, the mixture was stirred at room temperature for 3 h, following which the reaction medium was poured into cold water (100 mL). The organic phase was recovered, dried under anhydrous magnesium sulfate and the solvent evaporated under reduced pressure. The crude residue was chromatographed under silica gel with dichloromethane as a solvent to give 1.51 g (66%) of lactame **2** (JM-00266). In this synthesis, only the *trans* diastereoisomer was obtained [61].

C_25_H_20_N_2_S_2_O_3_; MP °C = 118–120 (Diethyl ether)^1^H-RMN (CDCl_3_): δ 2.44, s, 3H, CH_3_; 2.49, s, 3H, CH_3_; 5.26, d, 1H, H_a_; 5.33, d, 1H, H_b_ (*J*H_a_H_b_ = 2.4 Hz); 7.18, d, 2 aromH, (*J* = 8.4 Hz); 7.25–7.30, m, 4 aromH; 7.78, m, 2H, H4″-H5″; 7.88, m, 2H, H3″-H6″.^13^C-RMN (100.6 MHz, CDCl) 16.45 (CH_3_); 16.50 (CH_3_); 60.99 (Cb); 62.77 (Ca); 118.15 (2C); 123.85 (C3″–C6″); 126.63 (2C); 127.00 (2C); 127.91 (2C); 131.65 (C2″–C7″); 132.10 (C1); 134.14 (C4–C1′); 134.61 (C4″–C5″); 140.11 (C4′); 161.75 (C1″–C8″); 166.65 (Cc).SM (ESI) *m*/*z* (%): 461.6 [M + H]^+^IR (KBr, cm^−1^): 3064 (ν aromCH); 2974, 2922, 2835 (ν aliphCH); 1759, 1714 (ν C=O).

#### 4.1.3. Synthesis of 1-(Trifluoromethyphenyl)-4-(4-Méthylthiophenyl)-3-Phtaloylazetidin-2-One (JM-00252)

This compound was synthesized according to the previously described procedure, with suitable starting materials. Compound JM-00252 was obtained with a good yield (72%).

C_25_H_17_F_3_N_2_SO_3_; Mr = 482,48; MP °C = 140 (Diethyl ether).^1^H-RMN (CDCl_3_): δ: 2.49, s, 3H; 5.38, d, 1H, *J* = 2.6 Hz; 5.31, d, 1H, *J* = 2.6 Hz; 7.28–7.33, m, 4 aromH; 7.43, d, 2 aromH, *J* = 8.6 Hz; 7.54, d, 2 aromH, *J* = 8.6 Hz; 7.76–7.79, m, 4 aromH;SM (ESI) *m*/*z* (%): 483.5 [M + H]^+^IR (KBr, cm^−1^): 1730, 1778 (ν CO); 2924–2974 (ν aliphC-H); 3070 (ν aromC-H).

### 4.2. Drugs

Arachidonylethanolamide (anandamide, AEA) and Arachidonyl-2′-chloroethylamide hydrate (ACEA) were supplied by Sigma (Saint-Quentin-Fallavier, France), Rimonabant (SR141716) was purchased from Sanofi Aventis (Paris, France). All compounds were stored at 50 mg·mL^−1^ in 100% DMSO/Tween 80 (4/1; *v*/*v*) and diluted with physiological saline before intraperitoneal (i.p.) and intravenous injections. When administered orally, drugs were directly dissolved in olive oil except for brain/plasma ratios (B/P) experiments.

### 4.3. cAMP Assay

GloSensor™ cAMP assay designed by Promega (Charbonnières-les-Bains, France) was used to monitor activation of CB1R. HEK293T/17 cells obtained from the American Type Cell Culture Collection (LGC Promochem, Molsheim, France) were seeded at 30,000 cells per well on poly-D-lysine coated 96-well plates (BD Biosciences, Le Pont de Claix, France). They were grown in Dulbecco’s Modified Eagle Medium (DMEM, Waltham, MA, USA) containing 2 mM glutamax and 10% fetal bovine serum. Twenty-four hours later, cells were transiently co-transfected with 50 ng per well of pcDNA3-mouse CB1R (plasmid #13391 supplied by Addgene, Cambridge, MA, USA) which was a gift from Mary E. Abood (California Pacific Medical Center Research Institute, San Francisco, California) and pGlo™-22F cAMP plasmid-luciferase (100 ng per well) allowing these cells to produce light proportionally to intracellular CB1R-dependent cAMP level. Transfection was performed using the Fugene^®^HD transfection reagent (Promega) for 24 h according to the manufacturer’s instructions. Mock transfections were performed with the empty vector pcDNA3. Twenty-four hours after transfection, cells were equilibrated for 2 h with a CO_2_-independent medium (Gibco^®^, Life Technologies, Courtaboeuf, France) supplemented with the GloSensor™ cAMP reagent before adding tested compounds. When a steady-state basal signal was obtained, cells were incubated for 10 min with 1 µM forskolin (Sigma) at room temperature to enhance the intracellular basal cAMP level. Then increasing concentrations of JM-00266 (0.1 µM to 1 mM) were added in the reaction medium in the presence or absence of anandamide (0.1 µM to 1 µM) and luminescence was measured for 20 min with a Victor 3V plate reader (Perkin Elmer, Courtaboeuf, France). Rimonabant and anandamide were used as positive controls. When GloSensor™ cAMP assays were carried out in the presence of 0.1 µg·mL^−1^ pertussis toxin (Sigma), this compound was added in the incubation medium 24 h before incubation with forskolin. All assays were performed in duplicate and each experiment was repeated three times. Results were expressed as the percentage of forskolin-stimulated luciferase activity. The EC50 values were generated using the Sigma Plot software.

### 4.4. Animals and Diets

Ten- to twelve-week-old C57BL/6J male mice (JanvierLabs, Le Genest Saint Isle, France), 10–12 week-old global CB1R^−/−^ mice (generous gift from Dr. James Pickel, National Institute of Mental Health, Bethesda, MD, USA) and 250–300 g male Wistar rats (JanvierLabs) were housed individually on a 12/12 h light/dark schedule at 22–23 °C with ad libitum access to water and to a standard diet (STD; AO4; UAR, Epinay-sur-Orge, France).

### 4.5. Pharmacokinetic Experiments

For determination of plasma absorption profile and tissue distribution, wild type mice were given an i.p. or orally dose of JM-00266 and/or Rimonabant via gavage (10 mg·kg^−1^). At 1, 2, 4 and 8 h post-administration, animals were killed by cervical dislocation and blood was collected from the vena cava into tubes containing heparin and centrifuged immediately at 6500× *g* for 10 min at 4 °C. For determination of tissue distribution, brain, liver and adipose tissue were collected 4 h post-gavage. Plasma and tissue samples were directly extracted for further analysis by Liquid chromatography coupled to tandem mass spectrometry (LC-MS/MS).

### 4.6. Quantitation of JM-00266 and Rimonabant by LC-MS/MS

Tissues were homogenized with a mini-bead beater (BioSpec Products, Bartlesville, OK) in 10 volumes of acetonitrile containing 15 ng of internal standard, JM-00252. Blood samples were centrifuged at 6500× *g* during 10 min 4 °C and 5 µL of plasma were diluted in 10 volumes of acetonitrile containing 15 ng of internal standard. Then, each homogenate of tissue or plasma was centrifuged for 10 min at 6500× *g* and the supernatant was recovered and carefully transferred to new glass tubes. Finally, the extract was injected into a 1200 LC system coupled to a 6460-QqQ MS/MS system equipped with a JetStream ESI source (Agilent technologies, Les Ulis, France). Separation was achieved on a Poroshell 120 EC-C8 2.1 mm × 100 mm, 2.7 µm column (Agilent technologies) at a flow rate of 0.4 mL.min^−1^, 55 °C, with a linear gradient of (solvent A) water containing 1 mM ammonium formate and 0.2% formic acid and (solvent B) methanol containing 1 mM ammonium formate and 0.2% formic acid and as follows: 50% B for 1 min, up to 100% B in 4 min, and then maintained at 100% for 2 min. The acquisition was performed in positive Single Reaction Monitoring (SRM) mode (source temperature: 300 °C, nebulizer gas flow rate was 10 L·min^−1^, 15 psi, sheath gas flow 7 L·min^−1^, temperature 225 °C, capillary 4000 V, nozzle 1500 V). Transitions used were JM-00266 461.1→337.1 (frag 116 V, CE 13 V), JM-00252 483.1→359.1 (frag 118 V, CE 13 V) and Rimonabant 463→363 (frag 118 V, CE 35 V). Rimonabant and JM-00266 were semi-quantitated by calculating their relative response to JM-00252, used as an internal standard. Results are expressed as ng·g^−1^ for tissues and ng·mL^−1^ for plasma.

### 4.7. Brain Penetrance

ADME (Absorption, Distribution, Metabolism and Excretion) predictions for JM-00266 and Rimonabant were performed using the Online Server Swiss-ADME [62]. The physicochemical parameter “Topological Polar Surface Area” (TPSA) was used to predict the blood–brain barrier (BBB) permeability of the two compounds.

Brain/plasma ratios (B/P) were determined as previously described [30]. Briefly, wild type mice were subjected to an i.p. or oral administration of JM-00266 and/or Rimonabant (10 mg·kg^−1^) and were killed by cervical dislocation at maximal absorption time point previously determined by pharmacokinetics experiments. Blood and brain were collected and directly extracted for further analysis by LC-MS/MS.

The brain uptake index (BUI), a reference method used to determine brain penetration was determined in rats as previously described [30]. Rats were subjected to carotid perfusion for 1 min with an equal quantity of JM-00266 and Rimonabant (1.5 mg) solubilized in DMSO/Tween 80 and diluted in a thermostated buffered solution (1.28 mM NaCl, 24 mM NaHCO_3_, 4.2 mM KCl, 2.4 mM NaH_2_PO_4_, 1.5 mM CaCl_2_ et 0.9 mM MgCl_2,_ 5 mM D-Glucose). The heart ventricles were severed before starting perfusion to avoid blood recirculation in the periphery. After 1 min, rats were killed and the brain was removed and directly extracted for further analysis by LC-MS/MS. BUI was calculated using Rimonabant as a reference compound as it is known to easily penetrate the BBB.

### 4.8. Body Temperature Measurement

Mice rectal temperature was assessed using a VT4810 controller (Vertex Technology Corp., New Taipei City, Taiwan) adapted with a mouse rectal probe. In the first experiment, the impact of anandamide (10 mg·kg^−1^) on body temperature was recorded during 30 min on animals injected either 20 min earlier with vehicle or JM-00266 (20 mg·kg^−1^). In a second experiment, mice were injected either with the vehicle, Rimonabant (10 mg·kg^−1^) or JM-00266 (20 mg·kg^−1^) 20 min before receiving the selective CB1R agonist ACEA (1 mg·kg^−1^) and rectal temperature was recorded 30 min later.

### 4.9. Behavioral Tests

For food intake experiments, mice were food-deprived for 24 h before being submitted to an i.p. injection of ACEA (1 mg·kg^−1^). Twenty minutes later, mice were injected with either vehicle, Rimonabant (10 mg·kg^−1^) or JM-00266 (20 mg·kg^−1^) and given free access to a pre-weighed amount of standard chow. Cumulative food intake was measured every hour over a period of 3 h from refeeding.

Anxiety-related behavior was assessed using the open field test (OF; Mouse Open Field Arena ENV-510; Med Associates Inc., Fairfax, VT, USA). Thirty min after an i.p. injection (10 mg·kg^−1^) of Rimonabant, JM-00266 or vehicle, animals were allowed for a 10 min free exploration in the plexiglass test chamber (43 cm × 43 cm) equipped with infrared photocell beams allowing horizontal activity tracking (total ambulatory distance, time and number of entries in the center) using the Activity Monitor software (SOF-811; Med associates Inc.). The two regions were defined by grid lines that divided each box into the center and periphery, with each of the four lines being 11 cm from each wall.

### 4.10. Gastro-Intestinal Transit

Short-term effects of JM-00266 on gastro-intestinal transit were tested as an indicator of peripheral CB1R activity. Transit was measured by administering a non-absorbed marker (10% charcoal suspension in 5% gum arabic) as previously described [27].

### 4.11. Oral Glucose and Insulin Tolerance Tests

To test acute effects of JM-00266 on glucose metabolism and their specificity to CB1R, C57BL/6 wild type mice and CB1R^−/−^ mice fasted for 6 h were subjected to an oral glucose tolerance test (OGTT, 2 g·kg^−1^ glucose) and to an insulin tolerance test (ITT) using 0.5 UI·kg^−1^ insulin (Actrapid, Novo Nordisk, La Défense, France) 10 min after i.p. injection of JM-00266 (10 mg·kg^−1^) or vehicle. For both tests, glycemia was monitored during 120 min post-injection with a My Life Pura TM glucose meter (Ypsomed, Paris, France) directly on a blood drop from the tail. Corresponding AUC_0–2 h_ were calculated.

### 4.12. Lipolysis Experiments

In vivo experiments. Overnight-fasted mice were injected i.p. with JM-00266 (mg·kg^−1^) or vehicle. Four hours later, mice were injected i.p. with the specific ß3-adrenergic receptor agonist, BRL37344 (5 mg·kg^−1^). Blood samples (~25 µL) were collected from tails in tubes containing EDTA (Sarstedt, Nümbrecht, Germany) at times 0, 15, 30, 45 and 60 min after BRL37344 injection to measure plasma glycerol appearance using a colorimetric assay (glycerol FS, DiaSys, Condom, France).

In vitro experiments. Overnight-fasted mice were injected i.p. with JM-00266 (mg·kg^−1^) or vehicle. Four hours later, mice were anesthetized with pentobarbital (50 mg·kg^−1^) and epididymal adipose tissue was collected, rinsed in a warm medium (DMEM-HAM/F12) supplemented with 1% BSA, and dissected into ~10 mg pieces. Then, 5–6 explants were incubated in a 5% CO_2_ atmosphere at 37 °C under slight agitation in 1 mL of the same medium. Free glycerol in the culture medium was determined after 1 h of incubation using a colorimetric assay (glycerol FS, DiaSys).

### 4.13. Statistical Analysis

Results are expressed as means ± SEM. Data were analyzed statistically with GraphPad Prism software (GraphPad Software, Inc., La Jolla, CA, USA) using 2-way ANOVA followed by the Tukey posthoc test or using the Student’s t-test. Differences were considered significant at *p* < 0.05.

## 5. Patents

The structure of JM-00266 was claimed in the United States and international patents US20180265498 and PCT/EP2016/072735. Preparation of 1,4-bis(4-methylphenyl)-3-phtaloylazetidin-2-ones and derivatives thereof useful for the treatment of diseases associated with hyperactivity of the endocannabinoid system. Robert, J.; Troy-Fioramonti, S.; Demizieux, L.; Degrace, P.

## Figures and Tables

**Figure 1 ijms-23-02923-f001:**
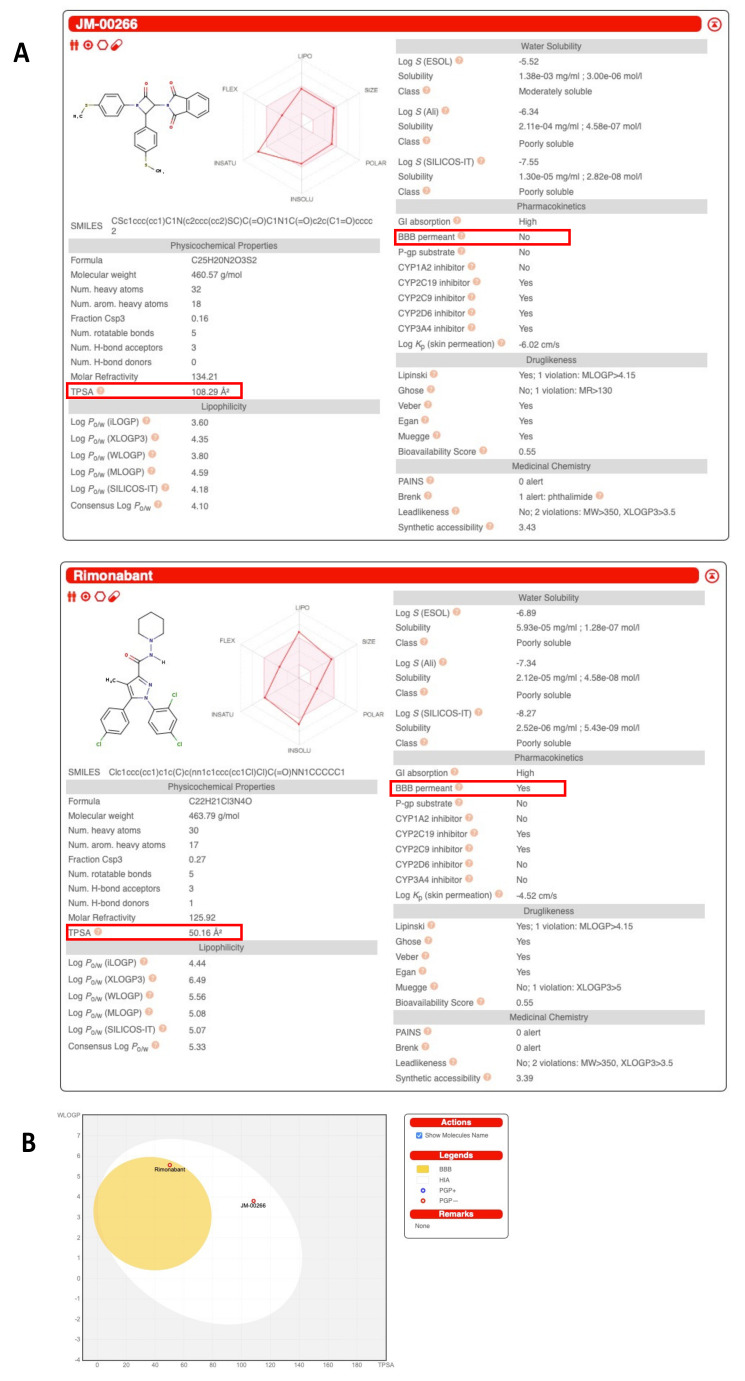
Structure and Swiss ADME analysis of JM-00266 in comparison with Rimonabant. (**A**): Chemical structures of JM-00266 and Rimonabant and results of their respective analysis using the Swiss ADME server. Boxes indicate topological polar surface area (TPSA) and potential ability to cross blood–brain barrier (BBB). (**B**): WLOGP (lipophilicity) vs. TPSA (apparent polarity) boiled-egg view of JM-00266 in comparison with Rimonabant.

**Figure 2 ijms-23-02923-f002:**
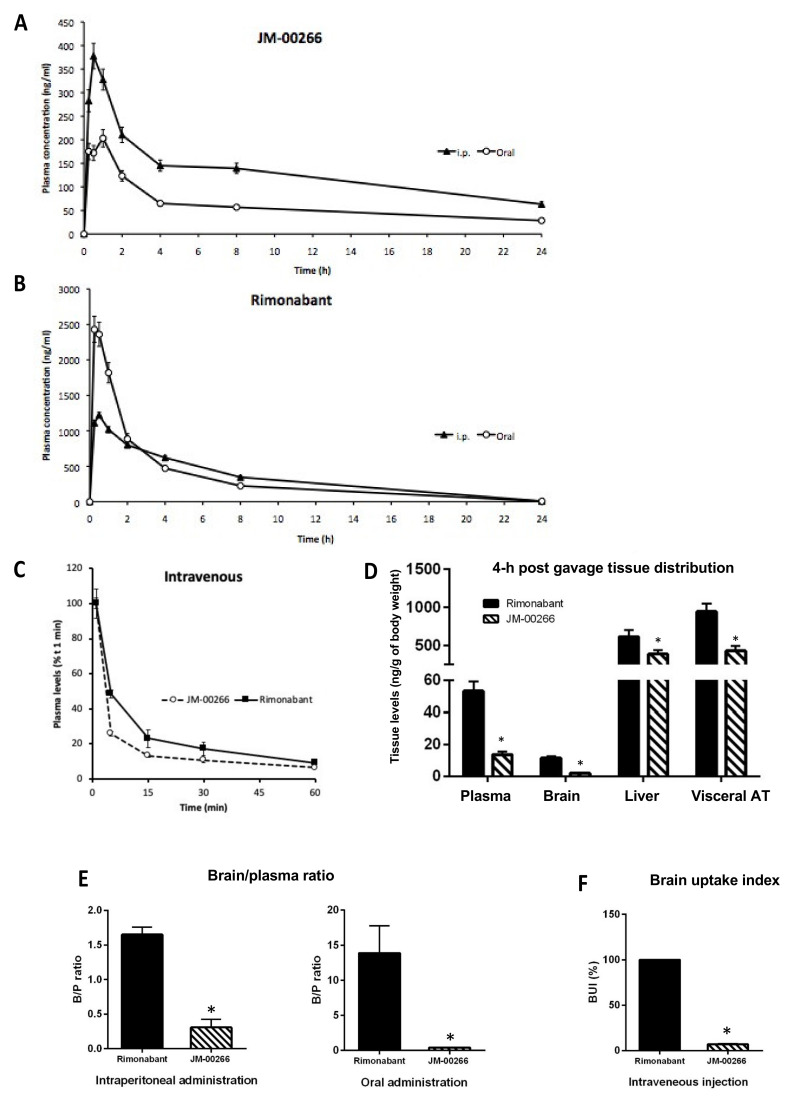
Plasma kinetic profile, tissue distribution and brain penetrance of JM-00266. (**A**,**B**): JM-00266 and Rimonabant plasma appearance after intraperitoneal and oral administration (10 mg·kg^−1^). Drugs were dissolved in DMSO/Tween 80 (4/1; *v*/*v*) and diluted with physiological saline before intraperitoneal injection. Results are expressed as mean ± SEM (*n* = 4 per group). (**C**): Plasma clearance of JM-00266 and Rimonabant after intravenous administration (**D**): JM-00266 and Rimonabant partitioning in plasma, brain, liver and visceral adipose tissue after oral administration (10 mg·kg^−1^). Results expressed as ng·g^−1^ of body weight were obtained multiplying concentration in sample by total tissue mass and normalizing by body weight (*n* = 4 per group) * *p* < 0.05. (**E**): Brain/plasma ratio after intraperitoneal or oral administration of JM-00266 and Rimonabant (10 mg·kg^−1^) to mice. Results are expressed as mean ± SEM (*n* = 4 per group) * *p* < 0.05. (**F**): Brain uptake index was determined in rats after intravenous co-administration of JM-00266 and Rimonabant (10 mg·kg^−1^). Results are expressed as mean ± SEM (*n* = 4 per group). * *p* < 0.05. In all experiments, drugs were dissolved in DMSO/Tween 80 (4/1; *v*/*v*) and diluted in a thermostated-buffered solution for intraperitoneal and intravenous injections or dissolved in oil for oral administration.

**Figure 3 ijms-23-02923-f003:**
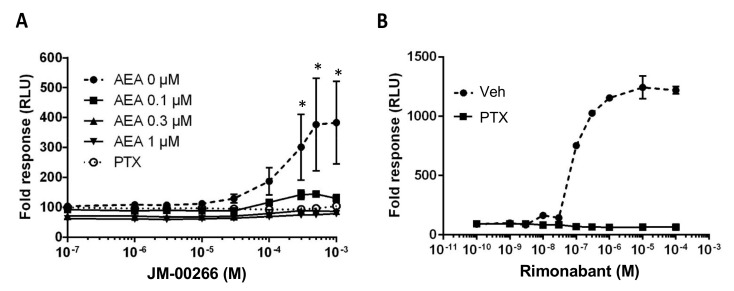
cAMP functional assay. Effect of increasing concentrations of JM-00266 (**A**) or Rimonabant (**B**) on cAMP Glosensor luminescence in the absence or presence of anandamide (AEA; 0–1 µM). When used, pertussis toxin (PTX; 0.1 µg·mL^−1^) was added 24 h before the cAMP assay. Intracellular cAMP levels were expressed as % variation of relative luminescence units (RLU) obtained after stimulation by forskolin (1 µM) for 10 min (*, *p* < 0.05).

**Figure 4 ijms-23-02923-f004:**
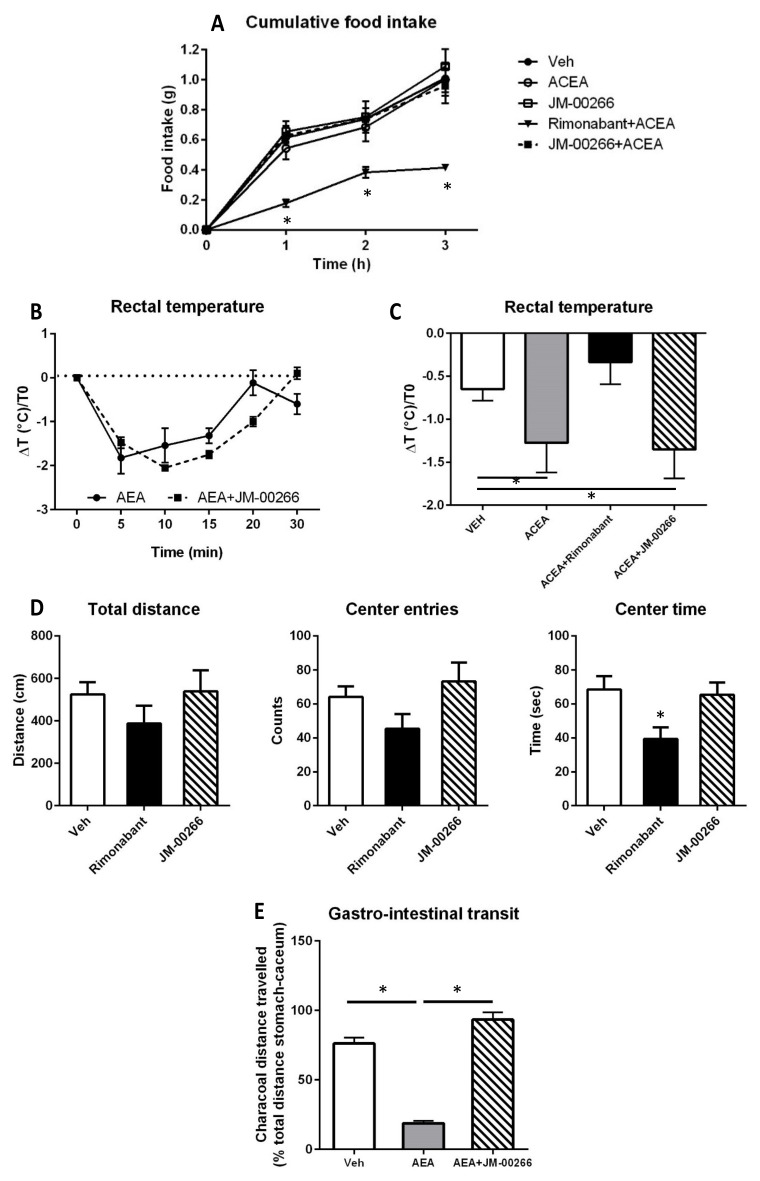
Effect of JM-00266 on behavior, body temperature and gastrointestinal transit. (**A**): Cumulative 3-h food intake in 24-h food-deprived mice subjected to an intraperitoneal injection of ACEA (1 mg/kg) alone, in combination with Rimonabant (10 mg·kg^−1^) or with JM-00266 (20 mg·kg^−1^). Results are expressed as mean ± SEM (*n* = 4 per group), * *p* < 0.05. (**B**): Effect of AEA (10 mg·kg^−1^) on rectal temperature in mice treated with JM-00266 or vehicle (20 mg·kg^−1^). Results are expressed as mean ± SEM (*n* = 4 per group) * *p* < 0.05. (**C**): Effect of ACEA on rectal temperature measured at t = 30 min after injection of ACEA (1 mg·kg^−1^) in mice treated with JM-00266 (20 mg·kg^−1^) 20 min earlier. Rimonabant (10 mg·kg^−1^) or vehicle. Results are expressed as mean ± SEM (*n* = 4 per group) * *p* < 0.05. (**D**): Effect of JM-00266 and Rimonabant on open field exploration. Total ambulatory distance was determined calculating total distance travelled for 10 min. Anxiety was evaluated monitoring total entries in the center of the arena and total time spent in the center on a 10-min period. Results are expressed as mean ± SEM (*n* = 5 per group). (**E**): Effect of JM-00266 on gastrointestinal motility. Mice were treated with JM-00266 (10 mg·kg^−1^) or vehicle 10 min before an injection of anandamide (10 mg·kg^−1^). Results are expressed as mean ± SEM (*n* = 3 per group) * *p* < 0.05.

**Figure 5 ijms-23-02923-f005:**
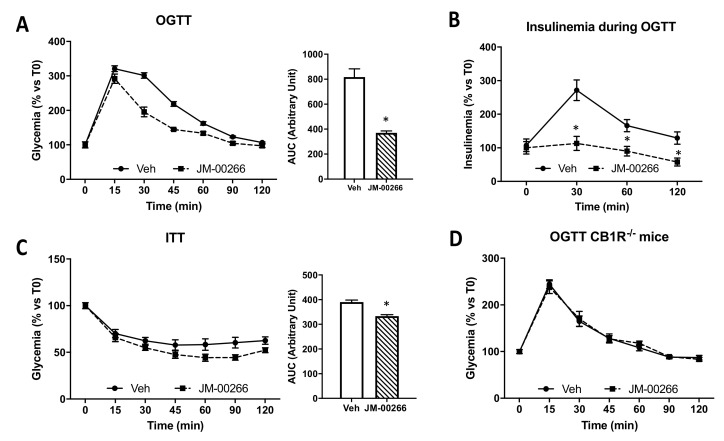
Effect of acute administration of JM-00266 on carbohydrate metabolism. (**A**): Oral glucose tolerance test (OGTT) and corresponding AUC_0–2 h_ calculations in mice injected with JM-00266 (10 mg·kg^−1^) or vehicle 10 min before glucose gavage (*n* = 10 per group). (**B**): Insulinemia during OGTT. (**C**): Insulin tolerance test (ITT) and corresponding AUC_0–2 h_ calculations in mice treated with JM-00266 (10 mg·kg^−1^) or vehicle 10 min before insulin administration (*n* = 15 per group). (**D**): OGTT in CB1R^−/−^ and corresponding AUC_0–2 h_ calculations mice injected with JM-00266 (10 mg·kg^−1^) or vehicle 10 min before glucose gavage (*n* = 7 per group). Results are expressed as mean ± SEM, * *p* < 0.05.

**Figure 6 ijms-23-02923-f006:**
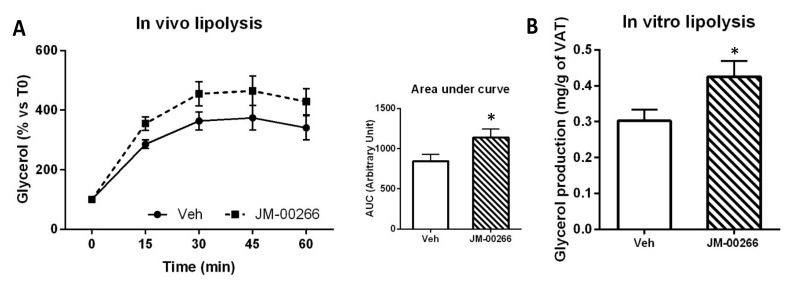
Effect of JM-00266 on adipose tissue lipolysis. (**A**): Plasma glycerol appearance and corresponding AUC_0–2 h_ calculations in mice treated with JM-00266 or vehicle 4 h before i.p. injection of BRL37344 (ß3-adrenergic receptor agonist, 5 mg·kg^−1^). Results are expressed as mean ± SEM (*n* = 5 per group), * *p* < 0.05. (**B**): Glycerol produced by visceral adipose tissue explants prepared from animals (*n* = 4 per group) was injected 4 h earlier with a dose of JM-00266 (10 mg·kg^−1^) or vehicle and cultured in DMEM-HAM/F12 medium supplemented with 1% BSA for 1 h. Results are expressed as mean ± SEM, * *p* < 0.05.

## Data Availability

The data presented in this study are available on request from the corresponding author.

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
