# Peer review of "Chemical Synthesis, Pharmacokinetic Properties and Biological Effects of JM-00266, a Putative Non-Brain Penetrant Cannabinoid Receptor 1 Inverse Agonist"

_ijms, 2022, doi:10.3390/ijms23062923_

Round 1

Reviewer 1 Report

typing error in line 79 the chemical name of JM-00266 is not correct. The methylthio groups are missing

You show very interesting results, having a compound with low potential for BBB penetration. However, I have observed some discrepancies that need clarification. The compound JM-00266 has poor solubility: 3 microM/L. Still you measure an EC50 of 248.6 microM. That cannot be correct as such a high concentration in the aqueous phase cannot be obtained. Please explain.

The same applies actually to the PK experiments. The highest plasma concentration is obtained in the ip experiments and is in the range of 1 microM. If the EC50 on the CB1R is 248.6microM there cannot be an effect. Even taken into account that the PK studies were done with 10mg/kg and the functional studies with either 10 or 20 mg/kg. Please explain.

So, I cannot understand how the inverse agonistic activity of JM-00266 can explain the very interesting in vivo results you have obtained. Nor is it easy to understand how the EC50 on CB1R was obtained with such a poorly soluble compound.

Author Response

We thank the reviewer for his comments and we tried to address them the best we could.

First, we have appropriately modified the chemical name of JM-00266.

Secondly, we understand that the reviewer is concerned by solubility issues. JM-00266 water solubility is low as predicted by SwissADME. Even if the solubility was clearly increased by DMSO in our experiments it is indeed very unlikely that it reached 250 microM. It is however possible that the IC50 value obtained was overestimated because of the poor solubility of the drug when tested for functional assay with HEK cells as we could observe some visible signs of precipitation for the highest concentrations tested.

As for in vivo experiments, the drug solution administrated was clear showing no visible precipitate. However, our PK data suggest a lower bioavailability for JM-00266 compared to Rimonabant. This was likely due to a lower membrane permeation causing incomplete absorption by intestine (oral) or mesenteric vessels (i.p.).

However, the very important point is that, in spite of these limitations, JM-00266 shows activity on CB1R and induces in vivo biological effects. This led to the conclusion that this compound does have potential and our efforts should now focus on an improvement of the bioavailability of JM-00266 before assessing its potential to induce long-term beneficial effects on obese mice.

We will also test the possibility that JM-00266 acts as a pro-drug as it could be rapidly metabolized and be the source of active metabolites responsible for the effects observed in vivo.

Reviewer 2 Report

In the current study the authors focused on new derivatives for targeting peripheral CB1 receptors. These substances could be associate with anti-obesity potential and specific pharmacokinetics and pharmacodynamic properties have been examined. This study is well organized, and the authors present the relative results in a detailed and comprehensive way. These findings are interesting, and essentially contribute to issues related to endocannabinoid system, its peripheral activity and a new strategy for candidate drugs to treat obesity-related metabolic disorders.  

However, the authors should address some points.

Specific list is provided below:

  • Please clarify how these doses were chosen for JM-00266 kinetic studies. More doses should be presented in order to have a real picture of kinetic profile
  • Concerning in vivo experiments including evaluation of CNS effects, gastro-intestinal motility and metabolic profile please add more doses to have a better evaluation.
  • According to the relative literature the dose of Rimonabant used in the present study is rather high. Please enrich the discussion on the comparison of the results between JM-00266 and Rimonabant especially for CNS effects. The clarification of the concept based on the choice of the doses used for each substance is needed.
  • Please enrich the discussion related to bioavailability and potency issues of JM-00266. Could you provide some directions for increasing bioavailability on future studies?

Author Response

First, we thank you for the compliments regarding the manuscript. However, you raised some points that we attempted to address in the following sections:

Concerning the doses chosen: as JM-00266 was designed from the structure of the well-characterized CB1R antagonist Rimonabant, we used Rimonabant as comparison for most of the experiments. The dose of Rimonabant retained was 10 mg/kg because it is commonly used in studies dealing with the peripheral effect of the drug on metabolism in mice (Jbilo et al. 2005, Ravinet-Trillou et al. 2003, Cota et al. 2009, Gary-Bobo et al., Noguerias et al. 2008, Watanabe 2008 et al., Son et al. 2010, Jourdan et al. 2010…). In addition, this dose was shown to induce brain CB1R-dependent behavioral effects (Patel and Hillard 2006; Wiley et al. 2005). The dose of JM-00266 was consequently fixed at 10 mg/kg for PK comparison purposes. However, since data PK analysis revealed a lower potency of JM-00266 compared to Rimonabant, we decided to use a higher dose of JM-00266 (20 mg/kg) for behavioral and metabolic studies. This argumentation is now mentioned in the discussion section.

Besides, it is difficult to conceive other experiments to test higher doses because of solubility issues as were raised by reviewer 1.

Concerning the comparison of the results between JM-00266 and Rimonabant especially for CNS effects: As indicated earlier, the dose of 10 mg/kg was shown to induce brain CB1R-dependent behavioral effects in different studies (Patel and Hillard 2006; Wiley et al. 2005). It is true that it is generally the highest concentration tested in studies dealing with psychiatric effects, but on the other hand, this dose is more adapted for testing peripheral CB1R-dependent effects. Hence we chose 10 mg/kg for both Rimonabant and JM-00266 for comparison purposes.

We have enriched the discussion concerning JM-00266 vs Rimonabant CNS effects as suggested.

Round 2

Reviewer 1 Report

chemical name is still not correct. It should be 1,4-di-(4-méthylthiophenyl)-3-phthaloylazetidin-2-one). One "h" from phthalo etc is missing. It seems to me that the  structure in suppl file is wrong: it is not a phthalimide, which I believe it might be.

I do no see that my concerns are addressed: lack of correlation between EC50 and efficacy in an animal model. EC50 is far too high to explain the in vivo efficacy

Author Response

We thank the reviewer for spotting the errors in the chemical name of JM-00266 and in the structure of JM-00252. We apologize for this and have made the appropriate changes.

We are aware and do agree that there is a lack of correlation between EC50 and efficacy. We have since then double checked our data and there is no mistake in the calculation of the EC50. We do not have a proper explanation for this discrepancy other that trying to improve the compound bioavailability.

That’s why our future efforts will focus both on the improvement of the bioavailability of the compound and on the possibility it could be rapidly metabolized and be the source of active metabolites. 

Reviewer 2 Report

The authors have adequately addressed all comments suggetsed by the reviewer  

Author Response

We are pleased that the reviewer is satisfied with our response